Digital Cell Sorter (DCS): a cell type identification, anomaly detection, and Hopfield landscapes toolkit for single-cell transcriptomics

Domanskyi Sergii 1 domansk6@msu.edu
Hakansson Alex 2
Bertus Thomas J. 1
Paternostro Giovanni 2
http://orcid.org/0000-0003-2762-8562 Piermarocchi Carlo 1
1 Department of Physics and Astronomy, Michigan State University , East Lansing, MI , USA
2 Sanford Burnham Prebys Medical Discovery Institute , La Jolla, CA , USA
Qin Zhaohui
Electronic publication date: 2021 Jan 13
Publication date: 2021
Volume: 9
Electronic Location ID: e10670
Received 2020 Oct 9; Accepted 2020 Dec 8
Copyright: © 2021 Domanskyi et al.
Copyright year: 2021
Copyright holder: Domanskyi et al.
License: This is an open access article distributed under the terms of the Creative Commons Attribution License, which permits unrestricted use, distribution, reproduction and adaptation in any medium and for any purpose provided that it is properly attributed. For attribution, the original author(s), title, publication source (PeerJ) and either DOI or URL of the article must be cited.
License URL: https://creativecommons.org/licenses/by/4.0/

Keywords: Transcriptome analysis software, Single cell RNA sequencing, Hopfield classifier, Hopfield landscapes visualization, Automatic cell type identification, Consensus annotation, Anomaly detection

Funding: National Institutes of Health R01GM122085 This work was supported by the National Institutes of Health (No. R01GM122085). The funders had no role in study design, data collection and analysis, decision to publish, or preparation of the manuscript.

==============================
Motivation

Analysis of singe cell RNA sequencing (scRNA-seq) typically consists of different steps including quality control, batch correction, clustering, cell identification and characterization, and visualization. The amount of scRNA-seq data is growing extremely fast, and novel algorithmic approaches improving these steps are key to extract more biological information. Here, we introduce: (i) two methods for automatic cell type identification (i.e., without expert curator) based on a voting algorithm and a Hopfield classifier, (ii) a method for cell anomaly quantification based on isolation forest, and (iii) a tool for the visualization of cell phenotypic landscapes based on Hopfield energy-like functions. These new approaches are integrated in a software platform that includes many other state-of-the-art methodologies and provides a self-contained toolkit for scRNA-seq analysis.

Results

We present a suite of software elements for the analysis of scRNA-seq data. This Python-based open source software, Digital Cell Sorter (DCS), consists in an extensive toolkit of methods for scRNA-seq analysis. We illustrate the capability of the software using data from large datasets of peripheral blood mononuclear cells (PBMC), as well as plasma cells of bone marrow samples from healthy donors and multiple myeloma patients. We test the novel algorithms by evaluating their ability to deconvolve cell mixtures and detect small numbers of anomalous cells in PBMC data.

Availability

The DCS toolkit is available for download and installation through the Python Package Index (PyPI). The software can be deployed using the Python import function following installation. Source code is also available for download on Zenodo: DOI 10.5281/zenodo.2533377.

Supplementary information

Supplemental Materials are available at PeerJ online.

Introduction

Several platforms have emerged over the last decade for high-throughput RNA sequencing at the single-cell resolution (Zheng et al., 2020). Obtaining biological insights from single-cell data, however, requires complex computational analysis. Generally, the first part of the analysis includes quality control (QC) of raw base call (BCL) files and alignment of reads with the reference genome, followed by their quantification at the gene-level. The second part of the analysis starts with additional QC procedures to filter out cells with low library size and low number of unique genes sequenced, detect multiples (i.e., two or more cells are captured instead of one), and remove cells with high content of mitochondrial genes, which indicates broken cells (Ilicic et al., 2016). Data is then normalized to ensure that the gene expression levels are comparable between individual cells. Existing normalization methods (Ding, Zheng & Wang, 2017) have different strengths and weaknesses. It is therefore important that scRNA-seq software provides the flexibility to choose the best normalization strategy for a given dataset. After normalization, the most variable genes’ largest principal components from Principal Component Analysis (PCA) are often used for clustering and two-dimensional visualization. After clustering, biological insight is obtained using different methods, including differential expression, network analysis, enrichment analysis and cell type annotation.

For users, it is most desirable to have an extensive toolkit in one software package. Existing packages for scRNA-seq data include Cell ranger (Zheng et al., 2017), edgeR (Robinson, McCarthy & Smyth, 2010), DESeg2 (Love, Huber & Anders, 2014), Seurat (Stuart et al., 2019), Scanpy (Wolf, Angerer & Theis, 2018), but many others are available. The advantages of an extensive set of methods under one open-source software umbrella are smooth integration of the processing steps and control of the workflow. For example, Seurat is a powerful package implemented in R with large data integration capability and visualization, Scanpy is a Python-based software specifically designed for efficient processing of large single-cell transcriptomics datasets. In addition, commercial providers of single-cell platforms have developed software tools for low-level processing and visualization (Eisenstein, 2020), focusing on the first part of the analysis, that is, to generating the gene counts matrix.

Here, we join the ongoing effort to provide the community with user-friendly tools for the second part of single-cell transcriptomics analysis, starting with pre-processing and quality control and ending with cell type annotation, visualization, and analysis of the annotated clusters. In addition to state-of-the-art methods, our Digital Cell Sorter (DCS) platform introduces three new algorithms: (1) An enhanced version of our recently-developed algorithm for the automatic annotation of cell types, polled DCS (pDCS) (Domanskyi et al., 2019b), which uses a predefined set of markers to calculate a voting score and annotate cell clusters with cell type information and its statistical significance. The enhanced version of pDCS uses a marker-cell type matrix normalization method to account for markers that are known to be unexpressed in certain cell types. Also, in the new version, low quality scores are set to zero to reduce noise in cell type assignment. (2) A tool that provides a cell anomaly score using isolation forest, an algorithm (Liu, Ting & Zhou, 2008) for anomaly detection. While clustering is based on similarity, our cell anomaly score detects and quantifies the degree of heterogeneity within each cluster and is important, for instance, in the analysis of scRNA-seq from cancer samples. We also use this score to detect cells that are different than the majority of the other cells in a dataset, like in the case of anomalous circulating cells in blood. (3) A second algorithm for cell type identification based on Hopfield networks (Hopfield, 1982). Hopfield networks allows for a direct mapping of associative memory patterns, in this case patterns of gene expression, into dynamic attractor states of a recurrent neural network. This method has been successfully used in the classification of cancer subtypes (Szedlak, Paternostro & Piermarocchi, 2014; Maetschke & Ragan, 2014; Cantini & Caselle, 2019; Udyavar et al., 2017; Conforte et al., 2020). In our algorithm, we use cell type markers to define Hopfield attractors, and we let clusters of cells evolve to align with these attractors. The Hopfield network is integrated with an underlying biological gene–gene network, the Parsimonious Gene Correlation Network (PCN) (Care, Westhead & Tooze, 2019), to retain only biologically significant edges. This allows us to obtain interpretable information on the role of specific markers and their local connectivity in defining the different cell types. The method also defines an energy-like function that permits the visualization of the gene expression landscape and represents cell types as valleys associated to the different cell type attractors.

The different tools in the DCS platform can be combined for improved performance. For instance, we show how to combine the methods in (1) (pDCS) and (2) (Hopfield classifier) into a consensus annotation methodology that is more accurate compared to the methods used separately. Finally, we provide examples of the capabilities of DCS and its performance using data from large single cell transcriptomics datasets of peripheral blood mononuclear cells (PBMC), and bone marrow samples from healthy and multiple myeloma patients. Note that our cell annotation methods are knowledge-based classifiers, since they rely on pre-existing knowledge from cell type markers and do not require training data.

Methods

Functionality overview and toolkit structure

DCS functionalities include: (i) pre-processing (handling of missing values, removing all-zero genes and cells, converting gene index to a desired convention, normalization, log-transforming); (ii) quality control and batch effects correction; (iii) cells anomaly score quantification; (iv) dimensionality reduction and clustering; (v) cell type annotation; (vi) visualization, and (vii) post-processing analysis.

We classify our tools into three categories: primary processing tools, data query tools, and visualization tools. Primary processing tools consist of functions transforming input files into valid tables and translating gene names to a desired convention, general data pre-processing, dimensionality reduction, and clustering. Cell type identification and anomaly score calculations are also included in the processing tools. Data query Application Programming Interface (API) tools include functions for retrieving gene expression of one or many genes across all cells or in a subset of cells, extraction of new marker genes characteristic of a cell cluster, and other query-type functionalities. Visualization tools include two-dimensional projection, quality control histograms, marker expression projections, marker expression summaries, gene expression heatmaps, individual gene t-texts, cell types assignment matrices, cell types stacked barplots, anomaly scores projections, pDCS null distribution histograms, new markers plots, Sankey diagrams (a.k.a. river plot), and cell type markers summary diagrams.

The DCS package is implemented in Python 3 and compatible with a variety of modern operating systems. The source code is deposited at https://github.com/sdomanskyi/DigitalCellSorter. Each new DCS release is archived on Zenodo, and can be installed via PyPI or GitHub, for example, by running command pip install DigitalCellSorter in user’s terminal. The Sphinx-build interactive documentation is available at https://digital-cell-sorter.readthedocs.io and as appendix of the Supplemental Materials. Tutorials and detailed description of each module and function of DCS as well as installation instructions for specific platforms are included in the documentation.

Cell type annotation

After unsupervised clustering, DCS automatically assigns clusters to cell types without relying on an expert curator to interpret the data, or a training data with cells already labeled. DCS uses all the information available in a knowledgebase of characteristic markers for many cell types. While cell type identification by manual interpretation generally provides good results, DCS assures that all the available information, including the presence and absence of markers, is taken into account, and can automatically identify cell types in very large datasets. Moreover, DCS provides statistical significance of the assignments, labeling as unknown clusters for which there is not sufficient information to make an assignment, and provides multiple labels with their associated scores when the expression pattern is consistent with more than one cell type. In DCS, automatic cell type annotation can be obtained using two methods: a voting algorithm and an Hopfield recurrent network classifier.

Voting algorithm

The voting algorithm in DCS is based on an extensive revision of our polled Digital Cell Sorter method (Domanskyi et al., 2019b). Prior information on cell markers is encoded in a marker/cell type matrix Mkm where k is the cell type, and m is the marker gene. The element Mkm = ± 1 if m is an expressed/not expressed marker of cell type k. We will refer to these as positive and negative markers, respectively. Finally, if marker m is not used in determining cell type k the corresponding element of M is zero. We normalize M, separately for negative and positive markers, by the number of markers expressed in each cell type and then by the number of cell types expressing each marker. Thus, markers that are unique to a particular cell type will be automatically assigned a large weight. By retaining markers in M that are expressed in a given dataset X, we obtain a matrix M~. In Fig. S1 (bottom) we show an example of M~ for a PBMC dataset (Zheng et al. (2017)), after gene filtering and normalization, with dark green (rose) corresponding to unique positive (negative) markers.

We then build the marker/centroid matrix Ymc of the mean expression of marker m across all cells in cluster c. For each marker m, we use Ymc to compute all cluster centroids’ z-scores Zmc. The z-score matrix Zmc is transformed into the matrix Z~mc=1 if Zmc ≥ ζ and Z~mc=0 otherwise, for a given threshold ζ. The number of possible supporting markers decreases by increasing the value of the cutoff ζ, and this parameter has to be selected so that each cell type expected to be present in the dataset has a sufficient number of markers. Figure S1 (top) shows Ymc, calculated for the PBMC dataset, with darker blue corresponding to higher expression of markers, and stars denoting statistically significant markers, that is, markers with z-score larger than ζ. We have varied the parameter ζ in the range 0.1–1.5, and for the dataset in the figure, we chose ζ = 0.3. Finally, we compute the matrix of voting scores for each type-cluster pair (k, c) according to Vkc=∑mM~kmZ~mc.

To quantify the statistical significance of the voting scores and make the final assignment, we use a randomization method to calculate the statistical uncertainty associated to each type-cluster pair (k, c). We randomize the clusters by preserving their size and assigning to them cells randomly chosen from the whole dataset, and compute the voting scores for each random configuration. This randomization is performed n = 104 times, recording the voting matrix Vkc for each configuration of random clusters. This method accounts for cluster sizes, the overall gene expression distribution of the markers, and imbalances in the number of markers per cell type in estimating the uncertainty. The procedure provides distributions of voting results Pkc(Vkc) for a null model of random clusters.

We determine the z-scores, Λkc, of the voting results Vkc in the null distribution Pkc(Vkc) and assign the cell type according to Tc = argmaxk Λkc. All cells belonging to cluster c are thus identified as cell type Tc. In the example of Fig. S1, we show in the top panel how, after the cell type of each cluster has been assigned, we use green/red stars to indicate supporting/contradicting markers. The labels assigned to clusters are indicated on the left side of the top panel, and the total numbers of cell of each type and cluster are shown in the right panel.

DCS also includes an algorithm that consists in a modification of the above voting method, and is similar to a recently-proposed evidence-based cell-type identification algorithm (Shao et al., 2020). In this modification, we divide the voting scores Vkc by the maximum possible scores that each cell type could have if all positive makers and none of the negative markers were expressed, (1) Λkc=Vkc/∑m,Mkm>0Mkm.

We then assign the cell type according to Tc = argmaxk Λkc. The main difference with respect to the algorithm in Shao et al. (2020) is that we account for negative marker genes as well as marker weights. Both annotation methods introduced in this section can be combined to obtain a consensus score using the geometric mean of the corresponding Λkc.

We remark that, in contrast to the previous version of our algorithm (Domanskyi et al., 2019b), we now account for negative markers, that is, markers that should not be significantly expressed, in the cell type assignment. The normalization procedure has been modified accordingly. A marker can be in one of three states: supporting, contradicting, or neither supporting nor contradicting. Moreover, to discard low quality score, we now set the score to zero if the number of supporting markers is below 10% of all known markers for a given cell type.

Hopfield knowledge-based classifier

Cell type assignment with this algorithm is based on the idea that gene expression for different cell types can be represented as associative memories, and encoded as attractor states of the signaling dynamics in an underlying gene network. If one starts with cells in a cluster and let them evolve according to the dynamics defined by a set of interacting memory patterns, the overlap of the cluster configuration with the attractors can be used for cell type assignment.

Hopfield networks (Hopfield, 1982) are the simplest models of associative memories and are defined using N Boolean variables σi(t) evolving at integer time steps t. In our case these variables are associated with the expression of each gene. The initial state of each node (gene) takes one of two values, σi(t) = ±1 (over/underexpressed), based on the statistical significance of the average marker expression in each cluster, determined as a z-score above a threshold ζ = 0.3 across all clusters. In the canonical Hopfield model, a coupling matrix is constructed to store a set of p independent Boolean patterns ξiμ=±1 as point attractors, where i = 0,1,…,N − 1 is the node index and μ = 0,1,…,p − 1 is the pattern index. In the algorithm implemented in DCS, we build attractors ξ~iμ using our normalized marker cell type matrix, M~, detailed above. The negative elements of this matrix are set to zero, thus negative markers are not used in this method. The coupling matrix Jij defines the strength and sign of the signal sent from node j to node i and is defined by (2) Jij=AijN∑μνξ~iμ(Q−1)μνξ~jν,

where (3) Qμν=1N∑iξ~iμξ~iν,

is a matrix that reduces the effects of correlation in the attractors (Amit, Gutfreund & Sompolinsky, 1985a, 1985b), and Aij is the adjacency matrix of the underlying biological gene-gene interaction network. DCS currently uses a generic gene correlation network, the Parsimonious Gene Correlation Network (PCN) (Huang et al., 2018), which can be easily replaced with other gene networks. The underlying network defined by Aij effectively reduces interactions between the nodes of the Hopfield neural network to retain only biologically-justified gene-gene interactions.

The total field at node i at time t is given by (4) hi(t)=∑jJijσj(t)+hiext,

and the dynamical update rule is given by (5) σi(t+1)={+1withprobability(1+e−2hi(t)/T)−1−1otherwise,

where T is an effective temperature representing noise (not a physical temperature). Biologically, this noise represents the effect of different kinds of biochemical fluctuations in cells. Additionally, any node in the network that is being turned off, that is, the local field on a given node is such that it would switch from +1 to −1, is instead set to 0. This simple modification to the dynamical rule above makes the signaling dynamics dependent not only on the neighboring genes, but also on the gene state at the previous step, and the network becomes more diluted during the evolution. Moreover, this way of updating the system is asymmetric since it affects only genes that switch from +1 to −1, and not, for instance, nodes that stay from -1 to −1. This modified Hopfield dynamics has been recently proposed by Cantini & Caselle (2019), and we have found, like they did, that this rule greatly improves the convergence of the classifier. This modification is useful because nodes correctly overexpressed in the input and associated to a +1 get sometimes switched to −1 only due to the stochastic nature of the algorithm. Removing these nodes avoids the amplification of these wrong switches in the following updating steps.

The rule of σ update, Eq. (5), may be implemented in various ways. The following choices of update schemes were previously described in Domanskyi et al. (2019a). Specifically, the synchronous scheme updates the state of all the nodes in the system at every time step, but this is sensible only if the simulated system has a central pacemaker coordinating the activity of all nodes. A more appropriate choice for decentralized systems is the asynchronous scheme, in which the state of a randomly chosen subset of nodes is updated at each time step. Here, we use the asynchronous scheme with update probability for each node that linearly increases from 0.025 to 0.5 in the course 100 time steps, after which it is kept constant.

The overlap of the state vector σi(t) with the μth pattern in one of the attractor states is given by (6) mμ(t)=1N∑i,νσi(t)ξ~iν(Q−1)νμ,

where − 1≤ mμ(t) ≤ + 1. This overlap is similar to the one defined in Domanskyi et al. (2019a). The overlap measures the similarity between the gene expression of the different attractors and the simulated gene expression, and mμ(t) = + 1 means that there is perfect agreement between the simulated expression and attractor pattern μ. For each cluster we assign the cell type corresponding to the maximum non-negative overlap with the attractor states at the last time point. If for a given cluster there are no non-negative overlaps with any of the attractor states, then a “Unassigned” label is assigned. The stochastic nature of evolution of Hopfield network may lead to slightly different dynamics between independent realization, therefore the simulation is repeated multiple times and the most frequently assigned cell type is selected as a label for each cluster. Second, third, and other most frequent cells types are also recorded into the score matrix with their corresponding frequencies.

The two methods for cell assignment in the previous section and the Hopfield classifier can be combined in different ways to obtain a consensus approach using the geometric averages of the corresponding score matrices. All combination options are detailed in the DCS documentation.

Hopfield landscapes visualization

The Hopfield model introduced in the previous section can be used to define a quasi-energy function (7) E=−12∑i,jσiJijσj,

where σi is the Boolean variable describing gene up or down regulation, and the matrix J is defined using Eq. (2). Eq. (7) defines a Lyapunov function for the signaling dynamics in the gene network and allows us to build a Hopfield landscape in which the attractor states, that is, the different cell types, are the minima of a complex multi-dimensional phenotypic landscape. The quasi-energy in Eq. (7) can be defined with or without the matrices Aij and Qμν, and using different normalizations for the ξiμ. Therefore, our DCS software allows for different options for the landscape representation. This landscape can be explicitly visualized by starting at the equilibrium points corresponding to the attractor configurations and adding noise to sample their basins of attraction. The points sampled can then be represented in a 2D plot using principal components projections (Szedlak et al., 2017; Maetschke & Ragan, 2014; Fard et al., 2016; Taherian Fard & Ragan, 2017). An example of output from this DCS visualization tool is in Fig. 1, where the Aij was used (and not the Qμν) and the ξiμ were normalized as in the table M~ (See Fig. S2 for a landscape with Qμν). The figure shows the visualization of the phenotypic landscape of 14 different hematopoietic cell types, where the coordinates are the first two largest principal components of the point attractors, and the colors and contour lines reflect the Hopfield quasi-energy function of Eq. (7). These attractors were built using markers from Newman et al. (2015), modified to merge subtypes of B cells, CD4 T cells, NK cells, Macrophages, Dendritic cells and Mast cells, and their position in the landscape are indicated by stars. This visualization effectively uses the Hopfield quasi-energy to represent the matching/mismatching with the attractors in all dimensions. Note how for some cell types the basins of attraction are closer to each other than for others, and sometimes they overlap. For instance, Mast Cells, Eosinophils, and Plasma cells have overlapping basins, well separated from other cell types such as T cells or Macrophages. Additional considerations and analyses on Hopfield attractors and the role of the Q matrix are discussed in Supplemental Materials (Figs. S2 and S3).

Figure 1 Hopfield attractor landscape visualization.

The points are colored according to their Hopfield quasi-energy, Eq. (7).

Cell anomaly quantification

This module implements an algorithm that quantifies cell anomaly, that is, how much cells are different from other cells within the same dataset or within one or more clusters. Anomaly detection and can be interpreted as the opposite of clustering, which is based on similarity measures. The module for the quantification of anomaly is based on the Isolation Forest anomaly detection algorithm (Liu, Ting & Zhou, 2008, 2012).

The Isolation Forest algorithm isolates cells by randomly selecting a gene and a split value for its expression value. A schematic illustration of the algorithm is shown in Fig. 2. For the sake of illustration, consider a genome with two genes A and B only (along the two axes in Fig. 2A). Random partitioning is illustrated by the vertical and horizontal lines labeled with s1,…,9. At each step the cells are partitioned in two sets by choosing one of the two genes and a random threshold for the chosen gene’s expression. For a typical (blue) cell (Fig. 2A), the number of steps necessary to isolate the cell is larger than for an anomalous cell (red). This recursive partitioning can be described by a tree, as shown in Fig. 2B, where each internal node contains a gene and a threshold value that were used in that partitioning, whereas leaf nodes are the isolated cells. The partitioning is carried out until all cells are isolated, that is, no further partition is possible. The cell anomaly algorithm has a training and an evaluation stage. In the training stage, an ensemble of n = 100 isolation trees, called isolation forest, is generated using random sub-samples of ψ = 256 cells from the original dataset. Next, in the evaluation stage, for each cell i in the dataset, or in a subset of cells of interest, and for each tree j in the forest, a path length hij is derived by counting the number of edges from the root node of the tree to the node where the cell is isolated by following genes and thresholds stored in tree j. To compute anomaly score of cell i, path lengths hij obtained from all n trees in the isolation forest are averaged and normalized according to (8) si=−∑jhijc(ψ)n,

where c(ψ) is a normalization factor that accounts for the sub-sampling size ψ, giving a single score for the cell i, or a measure of anomaly. The algorithm’s pseudocode, and details about normalization factor, special conditions in the anomaly score computation and optimal selection of n and ψ are discussed in the original publication of the anomaly detection algorithm (Liu, Ting & Zhou, 2008). When a forest of random trees collectively results in shorter branches for particular cells, they are likely anomalous cells. Cell anomaly can be used to rank cells from more anomalous to less anomalous. Then the top-ranked cells can be further analyzed to investigate the biological difference from the other cells in the dataset. This module can be useful to identify small numbers of cells that may not be otherwise separated into a distinct cluster.

Figure 2 Schematics of how the isolation forest algorithm quantifies cell anomaly.

In this example we consider only two genes A and B. One of these two genes is randomly selected and data is partitioned based on a random threshold applied to the chosen gene’s expression (vertical and horizontal lines labeled with s1,…,9). Common cells, for example, c6 in (A), will be isolated with a larger number of partitions compared to anomalous cells, for example, c10. (B) Tree representation of the algorithm, where each internal node contains a gene and a threshold and represents a partition of the cells. Random partitioning produces shorter branches for anomalous cells (red) compared to common cells (blue).

Quality control

Figure S3 shows an example of output from the quality control module. Count depth, that is, number of reads per gene, and gene counts, that is, number of non-zero genes per cell, are evaluated for each cell. Cell with count depths and gene counts that are below 50% of the median of their respective distributions are tagged as low quality. The quality control parameters defaults can be overwritten by a software user, as different datasets may require adjustment of the cutoffs.

The DCS quality control procedure also accounts for the fraction of mitochondrial genes expressed in each cell, as shown in Fig. S3C. The fraction of mitochondrial genes cutoff is chosen at the abscissa interception with a line that passes through the median of the distribution and median plus 1.5 standard deviations. The list of human mitochondrial genes is taken directly from MitoCarta2.0: an updated inventory of mammalian mitochondrial proteins (Calvo, Clauser & Mootha, 2016).

The integration of DCS with other algorithms for batch correction, clustering, dimensionality reduction and other visualization tools is detailed in the Supplemental Materials (Figs. S4–S7).

Results

Cell identification in mixtures

We compared the performance of our previous pDCS method (Domanskyi et al., 2019b), with the new Hopfield classifier and the consensus annotation classifier described in Voting algorithm. To evaluate the algorithms’ performance, we randomly generated 100 mixtures of pure cell types, representing a gold standard, and evaluated the algorithms based on their automatic annotation. As gold standard, we used data from CD14 monocytes, CD19 B cells, CD34 cells, CD4 memory T cells, CD4 naive T cells, CD4 regulatory T cells, CD4 helper T cells, CD8 cytotoxic T cells, CD8 naive cytotoxic T cells and CD56 NK cells from a FACS-sorted PBMC dataset (Zheng et al., 2017), in addition to endothelial (SRS2397417, SRS4181127, SRS4181128, SRS4181129, SRS4181130) and epithelial cells (SRS2769050, SRS2769051) annotated in the PanglaoDB database (Franzén, Gan & Björkegren, 2019). The total number of cells used was 98,752, 53% of which are T cells. Each independent synthetic mixture contains on average 5,000 cells, chosen randomly from the 16 cell types listed above in random fractions. As the algorithms uses prior knowledge in the form of marker genes, we tested the algorithms using two marker genes tables: (i) CD Marker Handbook containing 11 main cell types and covering all 16 cell-types and sub-types present in the gold standard set (BD Biosciences, 2020), and (ii) CIBERSORT LM22 (Newman et al., 2015), modified to merge subsets of B cells, T cells, NK cells, Macrophages, Dendritic cells and Granulocytes. Marker table (ii) does not have marker information for CD34 cells, Endothelial cells or Epithelial cells, which are 25.6% of all the cells. To quantify each method’s performance we calculated a multi-class weighted F1 score, excluding unassigned cells, and took the median value over 100 independent random mixtures. We also calculated the median of the fraction of cells that had cell type unassigned for the same random mixtures. The results are shown in Table 1. Note how, for both marker tables, the new Hopfield classifier performed better than the original pDCS method and that the consensus annotation outperformed all other methods.

Table 1 Performance of annotation methods evaluated using mixtures of pure cell type populations.

Median multi-class F1 and percentage of unassigned cells (U) using two lists of markers as knowledge-base.

Method	pDCS	Hopfield	Consensus	
Metric	F1	U (%)	F1	U (%)	F1	U (%)	
CD Marker	0.836	6.0	0.867	16.7	0.947	21.6	
CIBERSORT	0.740	6.6	0.942	22.3	0.944	22.6	

A particular case of cell identification in a mixture is shown in Fig. 3. This figure shows how the platform can be used to analyze complex mixtures involving different tissues. Here we combined scRNA-seq data from: (1) CD138+/CD38+ cells from bone marrow samples obtained from hip replacement surgery (Ledergor et al., 2018) and (2) PBMC (Zheng et al., 2017). The samples from (1) are dominated by plasma cells due to the CD138+/CD38+ pre-selection by flow cytometry, while plasma cells should be relatively rare in the PBMC sample. We therefore expect our algorithm to label most of the cells from (1) as plasma cells, and identify other cell types in the cells coming from (2). Figure 3A shows the t-SNE layout of the mixture of 8,448 CD138/+CD38+ cells and 1,000 PBMCs randomly selected cells from the PBMC dataset, after QC and batch effect correction. Clusters annotated with cell type information are shown in Fig. 3B and the relative size of the clusters are shown in Fig. 3C, including cells that did not pass QC. Annotation was obtained using the CIBERSORT LM22 list as knowledge-base. Note how in this particular case the algorithm correctly identified 96% of the cells from (1) that passed QC as “Plasma cells”, while most of the remaining was assigned to “Monocytes”. Cells from (2) were mostly identified as T cells, which is the dominant cell type expected in dataset (2). The Sankey plot in panel (D) shows on the left side the clusters with their label and on the right side the batches from datasets (1) (labeled by “hip”) and (2). The thickness of the lines is proportional to the number of cells in the corresponding cluster-batch pair. Note how most of the cells from dataset (1) are linked to clusters labeled either as plasma cells or failed QC.

Figure 3 (A) Mixture of 8,448 CD138+CD38+ cells and 1,000 PBMCs on a t-SNE layout. Batch effects arising from different datasets, labeled by different colors, were removed with COMBAT. (B) Same layout as in (A) with clusters annotations. (C) Relative fractions of the annotated cell types, including cells that failed QC requirements. Such cells are excluded from plots in (A) and (B). (D) River plot showing labeled categories, from (B) and (C), and their memberships for each of the batches.

Performance with different marker lists

Since our voting and Hopfield-based algorithms depend on a list of marker genes, we have studied the dependance of cell type assignment on different lists of markers by comparing the labels assigned to a gold standard. In addition to the CIBERSORT and the CD Marker Handbook introduced above, we created a new list of markers obtained by differential expression analysis of data from a large recently-published single-cell study (Han et al., 2020), and we also used a manually-curated list of cell-type markers from PanglaoDB (Franzén, Gan & Björkegren, 2019). These cell marker lists differ in the number of markers per cell type, presence/absence of negative markers, and overlap of markers across cell types. More importantly, these lists are based on different labels, often corresponding to different degrees of type/subtype refinement. As a gold standard we used a set of 68,579 PBMC cells from Zheng et al. (2017). Figure 4 shows Sankey plots connecting cells annotated using the consensus annotation and one of the markers list (on the left side of each plot) with the annotation in the gold standard (right side of each plot). This visualization allows for a direct comparison of cells labeled using different degrees of type/sub-type refinement. The plot also explicitly indicates cells that were labeled as “Unassigned” and cells that did not pass QC. Overall, a comparison among these four plots indicates that the annotation is quite consistent.

Figure 4 Sankey diagrams for 68,579 PBMC annotated cells (taken as gold standard, Zheng et al. (2017)) using marker gene information from: (A) CIBERSORT LM22, (B) CD Marker Handbook, (C) HCL peripheral blood samples, (D) PanglaoDB.

The plots connect cells annotated using our consensus annotation with one of the four marker lists (on the left side of each plot) with the annotation in the gold standard (right side of each plot).

Validation of anomaly detection

We have designed an in silico experiment to validate and demonstrate the utility of the cell anomaly score introduced in Cell anomaly quantification. The experiment mimics a scenario in which a small number of circulating endothelial cells are present in peripheral blood, and evaluates the algorithm on its ability to detect these cells using their anomaly score. We selected 1,637 cells (T cells and Monocytes of PBMC) from SRS3363004 (GSM3169075) and 70 endothelial cells from SRS3822686 (GSM3402081) to mimic the presence of cells that are rare in blood. Both datasets were annotated in PanglaoDB and have been sequenced using Illumina HiSeq 2500 and 10× Chromium. The two datasets have similar sequencing depth, median number of expressed genes per cell and fraction of good quality cells, as detailed in Table S1.

We combined 1,637 PBMC and 20 endothelial cells randomly chosen out of the total 70. We then calculated the anomaly score of each cell after DCS normalization. The normalization and anomaly score calculation was repeated 100 times, each time with new randomly chosen 20 cells out of the 70 available cells. Receiver operating characteristic (ROC) curves were calculated for each of the 100 realizations with the cell anomaly rank as threshold. The average ROC curve is shown in Fig. 5A. The AUC is 0.929, indicating that the anomaly score can efficiently identify cells of a phenotype different than the majority of the other cells in the dataset. Next, instead of normal endothelial cells we used Merkel cell carcinoma cells (from SRA749327, SRS3693909) resulting in AUC of 100-fold average ROC 0.951, Fig. 5B, and Kaposi’s sarcoma cells (from SRA843432, SRS4322341) resulting in AUC of 100-fold average ROC 0.844, Fig. 5C. As a negative control, we replaced the 70 endothelial cells with 928 T cells from a bone marrow sample (from SRS3805245, GSM3396161) sequenced using Illumina HiSeq 3000 and 10× chromium, but otherwise very similar to the endothelial cells in sequencing depth and median number of expressed genes per cell. As before, we chose 20 out of the 928 T cells and combined them with 1637 blood cells. The ROC curve averaged over 100 random realizations average is shown in Fig. 5, curve (D). The AUC is 0.594, indicating that cells of similar phenotype get similar score despite being sequenced in different experiments. We have also used 5 and 10 cells instead of 20 in a set of additional in silico experiments to verify that the resulting AUC values are similar to the experiments with 20 cells detailed above.

Figure 5 ROC curves based on the anomaly score rank for 20 cells mixed in a set 1,637 PBMCs.

The panels show average ROC from 100 random realizations (solid line) and a random model (dased line): (A) endothelial cells, (B) Merkel cell carcinoma cells, (C) Kaposi’s sarcoma cells, and (D) bone marrow T cells.

To emphasize the difference between anomaly detection and clustering, we have quantified how close anomalous cells are to clusters of other cells. First we analyzed the normal PBMC cells to identify two clusters, L and M, with 1,231 lymphoid cells and 407 myeloid cells, respectively. Then, we combined the PBMC cells with 20 randomly chosen endothelial cells, hereon denoted as cluster O, and we normalized and projected the expression data on the principal components. Next, we calculated the inter- and intra-cluster euclidean distance for the clusters L, M and O, and derived the Silhouette scores for each of the clusters (defined as the difference between the nearest inter-cluster distance and intra-cluster distance, divided by the larger of the two) and the average Silhouette score across all cell in clusters L, M and O. Repeating this procedure 100 times, each time selecting a new set of cells O, we calculated the average measures, summarized in Table S2. The table has four sections where the small cluster O corresponds to the four cases in Fig. 5. The analysis shows that the O clusters always have to lowest silhouette score, indicating that clustering algorithms based on an euclidean distance will not separate the cells in O from the L and M cells.

Discussion

We have introduced DCS, a platform for the analysis of single cells transcriptomics that includes new methodologies for cell type identification, anomaly quantification, and visualization. The platform incorporates numerous state-of-the-art algorithms and provides a user-friendly pipeline, starting with pre-processing and quality control, and ending with cell type annotation and downstream analysis of the annotated data. DCS is an open source software that leverages on scikit-learn, scipy, numpy, pandas and other powerful python libraries and is optimized for an efficient processing of large data sets. We have provided examples showing implementations and functionalities, and more technical details are included in the software documentation. Overall, we expect that this platform will be highly valuable for the bioinformatics research community.

More specifically, the novel methodologies introduced in the platform are: (1) an enhanced version of our method for the automatic annotation of hematological cell types (pDCS). This new version uses a normalization method that accounts for markers that are known to be always unexpressed in certain cell types. (2) A tool to quantify the anomaly of cells based on isolation forest, a machine learning algorithm for anomaly detection. This algorithm detects cells that are different than the majority of other cells in a dataset, but that do not cluster together. (3) A new method for cell type identification based on a Hopfield network classifier. This method represents different cell types as attractors of the signaling dynamics in a gene network. A measured cell’s expression evolves according to this model, and its label is assigned by the attractor is converging into. We have shown how to combine different algorithms in our toolkit to obtain a consensus score for the cell annotation. DCS includes extensive visualization tools, including a tool for visualizing Hopfield’s landscapes, in which cell types are represented as minima of an energy-like function representing the cellular phenotypic landscape.

The automatic identification of cell types is an important component in the analysis of single cell data (Abdelaal et al., 2019). The performance of cell type annotation on FACS-sorted single cell was significantly improved when the enhanced pDCS algorithm was combined with the Hopfield classifier. This improvement is due to the fact that the Hopfield classifier labels as “unknown” all the gene expression patterns that do not converge to any of the attractors, reducing the chance of mislabeling. We have also found that a pre-existing biological network can be used to improve cell type identification. The network was integrated in our approach by retaining only edges in the Hopfield network that are present in a non-specific biological network (Care, Westhead & Tooze, 2019). We have demonstrated that different marker lists result in a consistent cell type annotation. Negative markers in addition to positive markers leads to better performance in the cell classification.

The anomaly detection module of DCS ranks cells from more anomalous to less anomalous and visualizes the results. To validate and demonstrate the utility of the cell anomaly score, we have presented in silico experiments that mimic a scenario in which a small number of anomalous cells are present in peripheral blood. We evaluated the algorithm on its ability to detect these cells from single cell transcriptomics data using their anomaly score. As anomalous cells we used normal endothelial cells, carcinoma cells, and sarcoma cells, mixed into PBMCs. We combined all PBMC and a small number of anomalous cells, normalized the mixed dataset and calculated the cell anomaly score. The AUC of the average ROC (cell anomaly rank used as threshold) curve for all three synthetic mixtures was high, indicating that the anomaly score could be useful to detect these unexpected cells in single cell transcriptomics datasets.

Future extensions of the platform will include the integration of markers lists for more cell types, and pre-existing networks that are tissue or disease specific. For instance, in the case of single cell data from cancer samples, the number of mutations can be so large that signaling networks can undergo significant rewiring, modifying the attractor landscape.

Supplemental Information

Supplemental Information 1 Digital Cell Sorter Supplementary Materials.

Click here for additional data file.

Supplemental Information 2 Digital Cell Sorter Documentation.

Click here for additional data file.

We thank Dr. Yuanfang Cai from Department of Computer Science at Drexel University for evaluating architecture and design of Digital Cell Sorter.

Additional Information and Declarations

Competing Interests

Author Contributions

Data Availability

Carlo Piermarocchi and Giovanni Paternostro own shares of Salgomed, Inc.

Sergii Domanskyi conceived and designed the experiments, performed the experiments, analyzed the data, prepared figures and/or tables, authored or reviewed drafts of the paper, and approved the final draft.

Alex Hakansson performed the experiments, analyzed the data, authored or reviewed drafts of the paper, and approved the final draft.

Thomas J. Bertus performed the experiments, analyzed the data, authored or reviewed drafts of the paper, and approved the final draft.

Giovanni Paternostro conceived and designed the experiments, authored or reviewed drafts of the paper, and approved the final draft.

Carlo Piermarocchi conceived and designed the experiments, authored or reviewed drafts of the paper, and approved the final draft.

The following information was supplied regarding data availability:

Digital Cell Sorter is available at GitHub: https://github.com/sdomanskyi/DigitalCellSorter.

Source Code is available at Zenodo:

Sergii Domanskyi, & wangjiayin1010. (2020, November 26). sdomanskyi/DigitalCellSorter: DigitalCellSorter (Version 1.3.7). Zenodo. DOI 10.5281/zenodo.4292550.

The documentation is available at DigitalCellSorter:

Domanskyi S., Szedlak A., Hawkins N.T., Wang J., Bertus T., Hakansson A., Paternostro G., Piermarocchi C. (2020). DigitalCellSorter’s documentation. Available at https://digital-cell-sorter.rtfd.io.

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
