# Peer review of "Digital Cell Sorter (DCS): a cell type identification, anomaly detection, and Hopfield landscapes toolkit for single-cell transcriptomics"

_PeerJ, doi:10.7717/peerj.10670_

## Round 0.1 · original submission · Minor Revisions

Please respond to reviewer 3's comments.

Reviewer 1 ·

Basic reporting

The manuscript presents a suite of software elements for the analysis of scRNA-seq data. It is clearly written with professional structures and figures.

Experimental design

The research question is well defined and the in silico experiment is carried out in a systematic way.

Validity of the findings

The performance of the platform the authors presented is supported by the provided data.

Reviewer 2 ·

Basic reporting

The manuscript written by Domanskyi et al., is well written and described with no flaws in english language.

Experimental design

This manuscript falls under the aims and scope of PeerJ. This manuscript is well defined the research question and the development of DCS toolkit would be beneficial for those performing ScRNA data analysis.

Validity of the findings

All underlying data have been provided, they are robust, statistically sound and controlled. Conclusions are well stated, linked to original research question.

Additional comments

Dear Authors,

Thank you for the well written and well described manuscript about DCS toolkit. I hope it will be highly usable for researchers performing single cell RNASeq data analysis.

Regards

Reviewer 3 ·

Basic reporting

1. Currently the users do not have an input file directly created from some of the more popular softwares used in the upstream analysis. This would require some additional handling of the files before it could be loaded onto DCS. If the compatibility between DCS and other softwares can be worked out it will just expand the user base for DCS.

2.The Github repository is well well documented. However it might help to have some tutorial/vignettes along to ease the learning curve for new users.

3. The text in the figures are too small to read. Also the grey text in sankey(river) plot makes it difficult to read it well in the grey background.

Experimental design

1. The cluster annotation is a very important step in the scRNAseq data analysis, both in terms of biological interpretation as well as evaluating the performance of the clustering. The authors here presented with a ‘voting algorithm’ the annotates clusters based on preferred marker. This methods mostly depends on a well defined maker database and assigns cell types based on absence/presence of marker genes. While this might work great with many cell types, it can get difficult when cell types with not not known specific and exclusive markers or cell states. Have the authors tested or considered the annotation based on marker identification (unbiased differential testing in one cluster vs the rest) and then using that list to identify cell types based on databases?


2. The authors also performed anomaly detection to find cells that are different from the majority of the states using isolation forest algorithm. This they also went on to show was not an effect of just technical variabilities due to experiment condition , technology or sequencing depth. Sometimes a cluster can be a mix of celltypes, this seems like a good step to verify that all the cells clustered together are indeed similar. However, clustering usually considers some significant PCs, but it could have more PCs that are not as significant but exists in the cell types (for e.g differentiation state, activity state etc). This could sometimes also be a factor of doublets or multiplets. The AUC is a good indicator of of the model. However, to differentiate between subtle differences in state type of same cell type vs. in a mixture of 2 very different cell types, it would help if there is some stat to attached to make sense of it. For e.g Silhouette values that tells how close a member is to its own cluster compared to other clusters, or some value to signify the strength/significance of this difference.

Validity of the findings

no comment

Additional comments

The authors have presented here a python toolkit for the analysis of single cell sequencing data which performs 1. automatic cluster annotation based on predefined markers, 2. Detects heterogeneity within a cluster identifying cell anomaly, 3. Cell type identification based on a neural network algorithm - Hopfield networks. Owing to the ever increasing generation of scRNA seq data and more researchers looking for tools that provide a suite of functions in an easy to use documentation- this seems like a good solution for some of the more important and time consuming step of cell type annotation.

---

## Round 0.2 · accepted · Accept

The reviewers' critiques have been sufficiently addressed.